# Phage Diving: An Exploration of the Carcharhinid Shark Epidermal Virome

**DOI:** 10.3390/v14091969

**Published:** 2022-09-05

**Authors:** Ryan D. Hesse, Michael Roach, Emma N. Kerr, Bhavya Papudeshi, Laís F. O. Lima, Asha Z. Goodman, Lisa Hoopes, Mark Scott, Lauren Meyer, Charlie Huveneers, Elizabeth A. Dinsdale

**Affiliations:** 1Flinders Accelerator for Microbiome Exploration, Flinders University, Surt Rd, Bedford Park, SA 5042, Australia; 2Department of Biological Sciences, San Diego State University, 5500 Campanile Dr, San Diego, CA 92182, USA; 3Georgia Aquarium, 225 Baker St NW, Atlanta, GA 30313, USA; 4Norfolk Island National Park, Mount Pitt Rd, Norfolk Island, QLD 2899, Australia

**Keywords:** carcharhinid, shark, epidermis, denticle, virome, bacteriophage, metagenomics

## Abstract

The epidermal microbiome is a critical element of marine organismal immunity, but the epidermal virome of marine organisms remains largely unexplored. The epidermis of sharks represents a unique viromic ecosystem. Sharks secrete a thin layer of mucus which harbors a diverse microbiome, while their hydrodynamic dermal denticles simultaneously repel environmental microbes. Here, we sampled the virome from the epidermis of three shark species in the family *Carcharhinidae*: the genetically and morphologically similar *Carcharhinus obscurus* (*n* = 6) and *Carcharhinus galapagensis* (*n* = 10) and the outgroup *Galeocerdo cuvier* (*n* = 15). Virome taxonomy was characterized using shotgun metagenomics and compared with a suite of multivariate analyses. All three sharks retain species-specific but highly similar epidermal viromes dominated by uncharacterized bacteriophages which vary slightly in proportional abundance within and among shark species. Intraspecific variation was lower among *C. galapagensis* than among *C. obscurus* and *G. cuvier.* Using both the annotated and unannotated reads, we were able to determine that the *Carcharhinus galapagensis* viromes were more similar to that of *G. cuvier* than they were to that of *C. obscurus*, suggesting that behavioral niche may be a more prominent driver of virome than host phylogeny.

## 1. Introduction

The microbiome is an integral part of metazoan immunity, providing an adaptable buffer against pathogens [1]. These buffers are critical in the digestive tract [2,3] and mucous membranes [4,5] where pathogenic transmission is high. The external microbiome is especially important for marine organisms [6,7,8,9], which often have more permeable integumentary systems and encounter a dense environmental pathogen load [10,11,12,13].

The external microbiome is critical for sharks in the family *Carcharhinidae* because their external surfaces are frequently abraded by bites [14,15] and reef structures [16] during mating and feeding behaviors, respectively. Despite a high rate of injury, these sharks display low rates of epidermal infection and relatively quick healing [16]. The fact that carcharhinids have experienced relatively few evolutionary changes over the last 10–50 million years [17] suggests that the immune mechanisms they have developed are highly effective.

Sharks secrete mucus from their epidermis, which is covered by a layer of hydrodynamic placoid scales called dermal denticles [18] that reduce settlement and biofilm formation by environmental microbes [19]. This allows sharks to retain a species-specific epidermal microbiome which differs from their surrounding water column [7,9,20]. Microbiome samples taken from shark epidermal wounds did not differ taxonomically from unblemished skin surfaces [21], which indicates that the epidermal microbiome is resistant to invasion.

While an investigation of the microbial drivers of shark external immunity is underway [21,22], the bacteriophage component remains an unresolved question in science. The epidermal viral and phage communities of marine vertebrates are not well explored; to the best of our knowledge there has only been one targeted investigation of the epidermal mucus virome of any marine vertebrate [23], and none for sharks. Bacteriophages provide a direct benefit to metazoan host immunity through piggyback-the-winner dynamics [4], which indirectly benefits the host by regulating the microbiome [24,25,26,27,28]. Thus, to fully understand the role of the microbiome in the health of their vertebrate hosts, the virome must also be investigated. Our current knowledge of gene function in non-human viruses is scant, so the investigation will begin with virome taxonomy.

Here, we explore the viromes of three members of one of the oldest extant vertebrate groups, the tiger shark, *Galeocerdo cuvier,* the dusky shark, *Carcharhinus obscurus,* and the Galapagos shark *Carcharhinus galapagensis. Galeocerdo cuvier* is the only extant member of an outgroup genus within the carcharhinid family, while *C. obscurus* and *C. galapagensis* are very closely related members of the *Carcharhinus* genus with similar morphologies [29] that differ slightly in their diet and spatial usage [30,31,32,33,34,35,36]. These three species frequently co-occur in tropical and subtropical islands [33,35], allowing for a direct comparison of viromes from sympatric individuals. In this study, we aim to observe how the taxonomy of shark epidermal viromes varies across shark species. We found the viromes were largely comprised of uncharacterized phages and the two most similar sharks had the least similar viromes.

## 2. Materials and Methods

### 2.1. Sample Collection

Epidermal virome samples were collected from three carcharhinid sharks, the tiger shark, *Galeocerdo cuvier* (*n* = 15), the dusky shark *Carcharhinus obscurus* (*n* = 6), and Galapagos shark *Carcharhinus galapagensis* (*n* = 10). Sharks were caught near Norfolk Island (29.04° S, 167.95° E), a remote island off the coast of eastern Australia, in the late austral summer of 2020 (Figure 1A). *Galeocerdo cuvier* were caught using a drifting drum line with a 16/0 circle hook. Once caught, sharks were brought to the boat, secured to the side of the boat with a tail-and-belly rope and turned up-side down to induce tonic immobility [37]. *Carcharhinus obscurus* and *C. galapagensis* were caught from the Kingston pier using a barbless hook on a hand line. Once landed, the sharks’ eyes and gills were covered with wet towels to reduce stress and fresh seawater was poured into the buccal cavity and gills to produce a mild narcotic effect [37]. Total length (TL) was measured, and sex was determined by the presence or absence of claspers. Roto tags were affixed to the dorsal fin (visible in Figure 1C) to enable the identification of individuals should they be recaptured. Virome samples were collected from the dorsal flank of sharks using a “super-sucker” blunt recapture syringe [7] (Figure 1B). Seawater was filtered through a tangential flow filter system to remove microbes, then loaded into the super-sucker chamber, flushed over the skin of the sharks, and recollected in the chamber along with the host’s epidermal microbes and viruses (Figure 1C). The microbial slurry from the super-suckers was filtered through a 0.22-micron Sterivex filter (Merck Millipore, Darmstadt, Germany) to remove the microbes and leave the virions as effluent, totaling approximately 100 mL of viral sample per individual shark. All sharks were promptly released after sampling and briefly monitored to ensure they swam away unharmed. The same filtration procedure was performed with 100 mL of sea water collected from the surface of the water column between the two sampling areas. The viral fractions were stored at 4 °C prior to processing, which took ~4 months.

### 2.2. Virome Processing and Sequencing

Viral samples were concentrated from ~100 mL to 1 mL using 20 mL 50 kDa conical ultrafiltration tubes spun at 2000 rcf. Ultrafiltration was used for viral concentration in lieu of ultracentrifugation and CsCl gradients for the sake of cost-effectiveness and simplicity [38]. DNA was extracted using the Norgen Phage DNA extraction kit (Norgen Biotek Corp., Thorold, ON, Canada) with added Proteinase K and DNase steps to increase yield and decrease contamination, respectively. Shotgun DNA libraries were prepared with the Swift Accel-NGS 2S Plus kit (Swift Biosciences, Ann Arbor, MI, USA) for the Illumina MiSeq (v3 chemistry–300 bp paired end reads) (Illumina, Inc., San Diego, CA, USA). Libraries were checked for quality using the Agilent Bioanalyzer 2100 (Agilent Technologies, Santa Clara, CA, USA) prior to sequencing.

### 2.3. Bioinformatics and Statistical Analysis

Reads were run through Prinseq++ [39] for quality control, with a minimum Q-score threshold of 25 and all “*n*” bases removed. The remaining reads were annotated using the Hecatomb pipeline [40], which uses the UniProt and UniClust amino acid databases to annotate individual reads to the closest taxon. The output was filtered for annotations with an e value of less than 10^−20^, which represents a relatively strict cutoff. Host contamination could not be removed explicitly because genomes of these sharks have not yet been sequenced, but all eukaryotic reads were effectively filtered out by Hecatomb. Read annotation data were used to compare proportional abundance of viral taxa across individual sharks and among shark species.

Congruence in virome community composition among shark species was assessed at three different taxonomic levels: family, genus, and species/strain. Due to the irregular nature of virus taxonomy, the informative power of viral genus is somewhat muddled. However, many phage strains are named based on the genus or family of bacteria they reportedly infect, which is a means of grouping strains at a level narrower than viral family and broader than strain. Thus, for the purpose of some of these analyses, phage strains reported to infect the same bacterial host genera were combined to form a larger quasi- “genus” level group termed the “host-genus” level.

Annotated sequence abundance data was normalized with a fourth-root transformation [41] and standardized to percent abundance per sample [42]. Community diversity was measured using Simpson’s diversity index, and patterns of community composition were assessed qualitatively with non-parametric multi-dimensional scaling (nMDS). These patterns were then quantified using a permutational MANOVA (PERMANOVA) analysis based on Bray-Curtis dissimilarity, with 10,000 substitutional iterations to account for the uneven sample sizes among species. A SIMPER analysis of similarity was performed to assess both the average similarity of the shark viromes to each other, and the contribution of each taxon to that score [43]. Statistical analyses were performed using R 4.0.5 and the PRIMER 7 software package. All Prinseq-filtered reads from the viromes were compared using Mash [44] with a kmer size of 20 and a sketch size of 10,000 to account for similarities in reads that could not be annotated.

## 3. Results

We obtained 35,893,289 reads from 31 sharks and the single water sample, of which 2,079,317 (~5.79%) were successfully annotated by the hecatomb pipeline above the prescribed e < 10^−21^ threshold. Individual samples ranged from 393,548 to 3,371,953 base pairs, with most falling between 700,000 and 1,500,000 base pairs (Appendix A), indicating the methodology was highly successful in obtaining viruses from a small quantity of shark mucus. The data were uploaded to the Short Read Archive for public access.

All three shark species supported a rich and highly diverse virome. The total richness of the dataset is approximately 2360 strains and each shark sample contained reads that mapped to between 600 and 1000 different strains (Figure 2). The *C. galapagensis* virome had the highest average richness and diversity, and *C. obscurus* had the most variation among samples. The water column sample had the highest richness, primarily driven by a group of proportionally scarce reads that mapped to a variety of cyanophages.

The majority of the annotated reads from the dataset mapped to prokaryotic viruses (70.5%), while less than 1% mapped to eukaryotic viruses, and approximately 29.4% were not classified to the taxonomic order level. Eukaryotic viruses mapped to a plurality of 26 families, including *Mimiviridae*, *Reoviridae*, *Marseilleviridae*, *Corticoviridae*, and *Phycodnaviridae*. Most of the annotated bacteriophage reads showed similarity to members of the order *Caudovirales* (99.3%), and those which could be annotated to the family level or lower mapped primarily to the families *Myoviridae* (46.4%), *Siphoviridae* (40.0%), and *Podoviridae* (9.8%), with sparse representation from 13 other phage families (Figure 3).

At all levels of viral taxonomy, the viromes from all three shark species were least similar to the water column sample. *Carcharhinus galapagensis* viromes had the least intraspecific variation of the three shark species, followed by *G. cuvier* and *C. obscurus* (Table 1), and *C. galapagensis* viromes displayed the tightest multidimensional scaling clusters across all taxonomic levels (Figure 4). The viromes of *C. galapagensis* and *C. obscurus* differed significantly in composition at all three levels of taxonomy, and *C. galapagensis* generally differed from *G. cuvier* to a degree that was only statistically significant at the strain level (Table 2). SIMPER analyses indicated that no single taxon contributed > 1% of the dissimilarity among samples at the strain level.

Because so few reads were annotated, we compared the whole virome that included the unannotated reads using Mash (Table 3). At this whole-virome level, all sharks were least similar to the water column sample. *Carcharhinus galapagensis* had the highest within-group similarity, and *G. cuvier* viromes were, on average, more similar to *C. galapagensis* viromes than to other *G. cuvier* viromes.

## 4. Discussion

The DNA virome assemblages of *Carcharhinus galapagensis*, *Carcharhinus obscurus*, and *Galeocerdo cuvier* are similar, dominated by bacteriophages with minimal number of eukaryotic viruses. The abundance of each phage varies slightly between individuals, but no single strain contributed >1% of dissimilarity among species, indicating that the compounding effect of those slight taxonomic variations leads to intraspecific and interspecific dispersion. Contrary to expectation, the two most closely related shark species exhibited the greatest difference in virome composition.

The microbiome of shark epidermis includes *Pseudomonas*, *Pseudoalteromonas*, and various cyanobacteria like *Synechococcus* as common constituents [7,20], and reads mapping to phages that infect those genera are well represented in the data. A majority of the annotated bacteriophage reads mapped to the families *Myoviridae*, *Siphoviridae*, *Podoviridae*, all of which are members of the order *Caudovirales.* These represent the most studied families of phages and are typically the most identified bacteriophage groups in marine organismal viromes [23,45,46]. Our limited capacity for viral annotation leaves a large fraction of the reads grouped into broader, less informative categories at the order or family level. There is a substantial amount of “dark matter” in viral databases [47], which makes accurate analysis of marine host-associated viromes especially difficult. Due to the bias in databases towards medically relevant phages, the most easily identifiable strains in this dataset are not likely to be the most abundant. Instead, many of these more abundant individual strains will be lumped in as “unclassified virus” or “unknown *Caudovirales*”, which obscures their effect on community composition. A similar bias appears at narrower taxonomic levels: various reads mapped to medically relevant phage strains including phages that infect pathogenic species of *Escherichia* and *Klebsiella.* Despite this skew, there are several groups prominent in this dataset that correlate with the marine microbiome.

A majority of the annotated reads from all viromes mapped to bacteriophages. The fact that <1% of all annotated reads mapped to eukaryotic viruses is consistent with veterinary records of shark infections, which are rarely of viral origin [48]. A majority of the eukaryotic viruses present in the shark mucus were in the families *Lavidaviridae*, *Mimiviridae*, *Reoviridae*, *Marseilleviridae*, and *Phycodnaviridae*, all of which are primarily known to infect algae, plants, fungi and protists rather than vertebrates. There is some evidence that some members of the families *Reoviridae* and *Marseilleviridae* infect vertebrates, but none are documented in the marine environment. The number of eukaryotic viruses in the shark viromes does not account for RNA viruses, which represent a majority of known eukaryotic viruses and a minority of known prokaryotic viruses [49]. The Sterivex filtration method excludes most viruses larger than 220 nm, which would likely affect more eukaryotic than prokaryotic viruses [50]. Despite these biases, a DNA bacteriophage fraction of 65–70% is substantially higher than the epidermal mucus of corals [45] or comparably large teleosts fish viromes [23]. Although both of those aforementioned studies included RNA viruses in their analysis, the ≤0.05% relative abundance of annotated eukaryotic viruses in these carcharhinid samples is unexpectedly low.

The low abundance of eukaryotic viruses in shark mucus relative to other pelagic fish may be related to behavioral differences, since the shark species sampled in this study are more solitary and do not form tight schools of individuals or swim in together like some pelagic fish, e.g., tuna (*Thunnus* spp.). The primary eukaryotic viruses in tuna mucus were circoviruses and unclassified members of the family *Circoviridae*, which includes strains that infect teleosts [51]. While the transmission of teleost-associated *Circoviridae* is not fully understood, the most studied circoviruses are commonly transmitted through fecal-oral routes [51], which is a more plausible mode of infection for schooling tuna than solitary sharks. The lack of prolonged side-to-side contact between pelagic sharks also drastically reduces the opportunity for transmission of viruses harbored in epidermal mucus in general. The morphological and physiological differences in the epidermis are also a factor, since the lower mucus output and hydrodynamic contours of shark skin are less conducive to acquiring microbiota from their environment [19].

The taxonomy of the virome appears to be most stable among *C. galapagensis* individuals, as they displayed the least within-group variation of the three sharks (Table 1, Table 2 and Table 3, Figure 4). Within-group variation increased for all sharks at narrower taxonomic levels, but *C. galapagensis* viromes were still the most similar at all levels despite higher average species richness and diversity. This suggests that the unique strains *C. galapagensis* harbors are held in common between individuals, and the combination of greater richness and less within-group variation is the factor separating *C. galapagensis* from the other sharks in the resemblance matrices. By the same token, the lack of significant dissimilarities between *C. obscurus* and *G. cuvier* was driven by high within-group variation rather than inherent congruence, supported by their dissimilarity in unannotated reads.

The taxonomic dissimilarity between *C. obscurus* and *C. galapagensis* viromes is an unexpected and intriguing result. Shark epidermal microbiomes display a stronger pattern of phylosymbiosis than teleost fish [9], which suggests that two sharks from the same genus would have more parity in the epidermal viromes than they would with an outgroup shark from the same family. *Carcharhinus galapagensis* and *C. obscurus* are so genetically similar that they are capable of hybridizing when populations overlap [29], and the epidermal structure is similar. In addition to genetic and physical similarities, the two shark species share spatial patterns of habitat usage, experience similar environments, and were caught in the same location, yet this was the only pairing of species that consistently displayed significant dissimilarity across taxonomic levels. *Galeocerdo cuvier,* however, has an entirely different denticle structure (Figure 5) and spatio-temporal niche [30,35], but their virome community was only significantly different from *C. galapagensis* at the highly variable strain level and did not significantly differ from *C. obscurus* at all taxonomic levels. The lack of difference between *G. cuvier* and *C. obscurus* is a result of within-group variation more than an indicator of true similarity, but the disparity in within-group variation and richness between *C. galapagensis* and *C. obscurus* is more signal than noise.

Denticle structure has a strong influence on the settlement and retention of biotic particles on the epidermis [19], but the taxonomic dissimilarity between *C. galapagensis* and *C. obscurus* viromes implies that virome community composition is influenced by biotically driven factors and not solely by epidermal habitat structure. For example, a difference in the chemical composition of *C. galapagensis* and *C. obscurus* epidermal mucus may drive differences in virome taxonomy. The mucus may contain a variety of antimicrobial compounds secreted by commensal bacteria, as is the case with various species of stingrays [52], which select for certain microbes and thereby select for their obligate phages. Diet can affect the composition of the microbiome [53], which may explain the similarity between *G. cuvier* and *C. galapagensis* as those species have more overlap in their dietary niches [31,33]. The drivers of epidermal virome composition are clearly numerous and interactive, and include biotic variables beyond host phylogeny and morphology, and thus future studies on the virome would benefit from the inclusion of physiological measures.

## Figures and Tables

**Figure 1 viruses-14-01969-f001:**
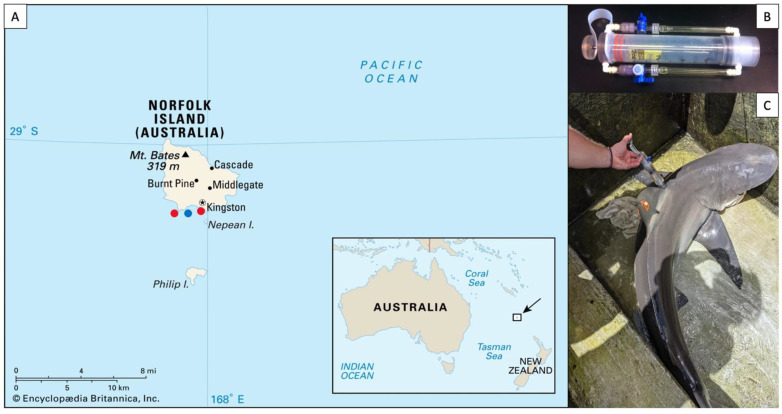
(**A**) Sampling locations on Norfolk Island, as indicated by red dots. The approximate location of the water sample is represented by a blue dot. (**B**) Photograph of a “super-sucker” microbiome sampling device. (**C**) Microbiome sampling on the dorsal flank of a *C. obscurus* individual.

**Figure 2 viruses-14-01969-f002:**
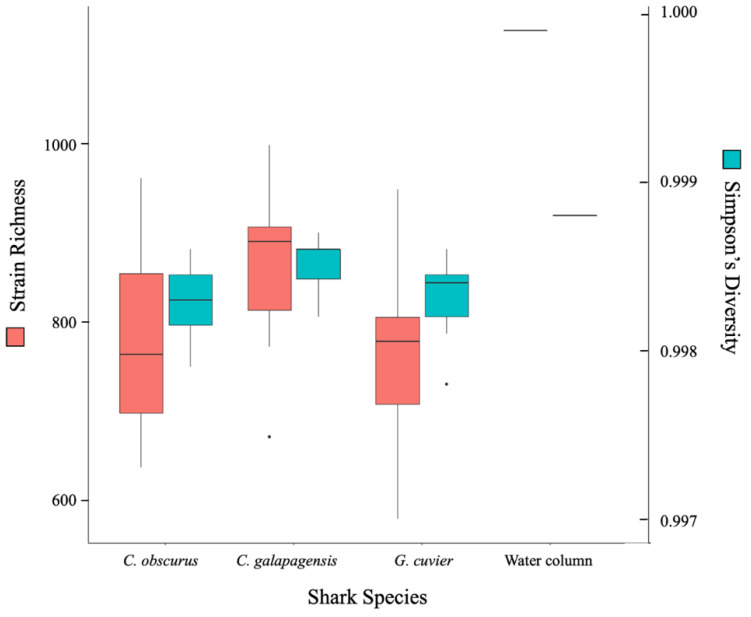
Box-and-whisker plot of richness (red) and Simpson’s diversity (blue) at the strain level for each shark species and the lone water column sample.

**Figure 3 viruses-14-01969-f003:**
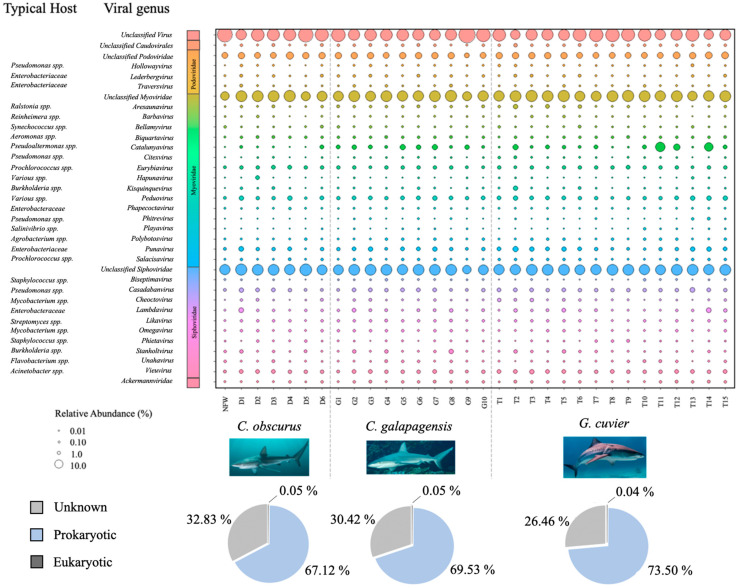
Bubble plot showing the relative abundances of all viral genera that contributed at least 0.5% of the total annotated reads for at least one virome sample. Larger bubbles denote greater abundance. The bacterial host taxa commonly associated with each genus (according to the NCBI taxonomy browser) are listed in the column to the left. Pie charts indicate the proportions of annotated viruses that infect eukaryotic (black), prokaryotic (blue), and unknown (grey) hosts for each shark species.

**Figure 4 viruses-14-01969-f004:**
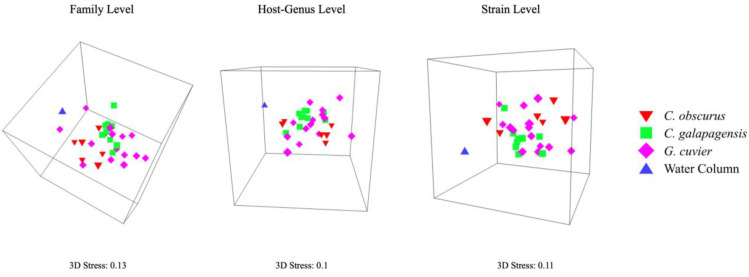
Three-dimensional nMDS plots showing the similarity of virome community composition at the family, host-genus, and strain levels. Plots are angled to reflect the separation of shark viromes from the lone water column sample, as well as the dispersion of shark viromes relative to each other.

**Figure 5 viruses-14-01969-f005:**
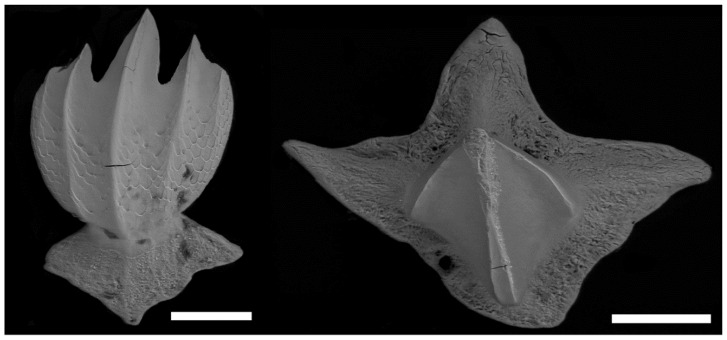
Electron microscopy images of the dermal denticles of *C. obscurus* (**left**) and *G. cuvier* (**right**) (photo by Erin Dillon and Jorge Ceballos). The scale bar represents a length of 100 μm.

**Table 1 viruses-14-01969-t001:** Average Bray-Curtis similarity of virome community composition among carcharhinid species and comparison to the water column sample at various levels of virus taxonomy.

Family Level	Water Column	*C. obscurus*	*C. galapagensis*	*G. cuvier*
*C. obscurus*	84.38	86.18		
*C. galapagensis*	84.55	86.99	90.35	
*G. cuvier*	82.21	86.09	88.13	86.63
**Host Genus level**	**Water Column**	* **C. obscurus** *	* **C. galapagensis** *	* **G. cuvier** *
*C. obscurus*	71.59	75.61		
*C. galapagensis*	73.64	77.39	81.88	
*G. cuvier*	70.73	76.61	79.02	77.33
**Strain level**	**Water Column**	* **C. obscurus** *	* **C. galapagensis** *	* **G. cuvier** *
*C. obscurus*	60.35	63.32		
*C. galapagensis*	63.07	65.57	71.97	
*G. cuvier*	58.90	64.20	67.64	65.17

**Table 2 viruses-14-01969-t002:** Summary of results for PERMANOVA and centroid distance analyses based on Bray–Curtis dissimilarity among carcharhinid species and the water column sample. Asterisks indicate *p*-values below the alpha threshold of 0.05.

Groups	t-Statistic	Unique Permutations	Distance Between Centroids	*p*-Value
Family Level				
Water Column, *C. obscurus*	1.21	7	12.88	0.14
Water Column, *C. galapagensis*	1.91	11	14.05	0.079
Water Column, *G. cuvier*	1.52	16	15.40	0.06
*C. obscurus*, *C. galapagensis*	1.68	5709	7.10	0.014 *
*C. obscurus, G. cuvier*	1.17	9058	5.54	0.47
*C. galapagensis*, *G. cuvier*	1.12	9910	4.01	0.085
Bacterial Host Genus Level				
Water Column, *C. obscurus*	1.28	7	24.05	0.15
Water Column, *C. galapagensis*	1.73	11	23.45	0.087
Water Column, *G. cuvier*	1.49	16	25.04	0.062
*C. obscurus*, *C. galapagensis*	1.44	5686	10.86	0.010 *
*C. obscurus*, *G. cuvier*	0.98	9045	7.80	0.49
*C. galapagensis*, *G. cuvier*	1.19	9877	7.27	0.093
Strain Level				
Water Column, *C. obscurus*	1.15	7	30.19	0.15
Water Column, *C. galapagensis*	1.52	11	30.45	0.096
Water Column, *G. cuvier*	1.31	16	31.44	0.13
*C. obscurus*, *C. galapagensis*	1.44	5652	15.82	0.003 *
*C. obscurus*, *G. cuvier*	1.04	9059	12.58	0.32
*C. galapagensis*, *G. cuvier*	1.22	9838	10.35	0.038 *

**Table 3 viruses-14-01969-t003:** The similarity of the 10,000 selected 20 mer hashes generated by Mash among shark species and the water column, shown as means with standard error.

Host	Water Column Percent Shared (SE)	*C. obscurus* Percent Shared (SE)	*C. galapagensis* Percent Shared (SE)	*G. cuvier* Percent Shared (SE)
*C. obscurus*	10.8 ± 1.1	12.3 ± 1.4		
*C. galapagensis*	12.2 ± 0.8	15.9 ± 0.7	29.6 ± 0.7	
*G. cuvier*	10.0 ± 0.9	13.2 ± 0.6	20.3 ± 0.7	15.8 ± 0.9

## Data Availability

Short Read Archive Accession for shark virome data: PRJNA850813.

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
