# Peer review of "Phage Diving: An Exploration of the Carcharhinid Shark Epidermal Virome"

_viruses, 2022, doi:10.3390/v14091969_

Round 1

Reviewer 1 Report

Shark epidermal viromes: species-specific and dominated by novel phages-Review

The present study provides a good snapshot of the phage diversity in the epidermal mucus of the three shark species analyzed. It sheds some light on the virome of wild animals, which still remains poorly analyzed so far. The analyses and their interpretation are correct, but some others should be carried out to have more robust results. I also think that the authors must reconsider the approach of the whole manuscript, as it gives a very detailed panorama of phage diversity but not of the whole virome present in the epidermis of sharks.

There are some comments that the authors should address:

Introduction

·       Please, add the size of the reference bars in Figure 1.

·       I would add a final sentence in this section to explain a little bit what the exact objective of the paper is and to understand why the authors tell the reader about the ecological niche of the three species of shark analyzed.

Material and methods

Sample collection

·       Please, add the permit/license for animal handling

·       As the authors sampled in two different locations, where did they take the sample named “water column”? Is it a mix of water samples from both locations? Please, clarify that.

Virome processing and sequencing

·       It is not clear what “kD” means in “20 mL 50 kD conical 121 ultrafiltration tubes”. Is it kDa?

Bioinformatics and Statistical Analysis

·       Reference in the third line of the first paragraph (Roach et al. 2022) is not cited properly in the text. Please, correct.

·       If the authors removed nucleotide-based alignments, I don’t think is necessary to write about them at all. The authors can tell that only amino acid alignments were used because they are more reliable when similarity with sequences in databases is not high.

·       Information about the similarity of their reads to those in databases is needed. This is key for the reader to know how possible it is that a read classified as a particular group of viruses actually belongs to that group. Also, it emphasizes that this kind of research is needed because they obtain viral sequences with low similarity to what is known so far. Finally, the authors may compare whether these similarity percentages vary significantly among the different shark species too.

The authors should be aware of that and specify in the results that the reads “showed similarity to” a specific group of phages as it cannot be assumed that they belong to that group for sure (especially in cases like this, where there are no other analyses supporting that).

·       Didn’t the authors obtain any complete genome or ORF? It would be very interesting to carry out some phylogenetic analyses to see how different the reads from this study are from the known viruses of their group.

Results

·       It should be written in the main text that the data was uploaded to the SRA database.

·       Table 1 should be in supplementary material. In any case, I would make it more attractive to readers by removing all lines between table rows.

·       The authors included Corticoviridae among the Eukaryotic viruses found, but they are, as far as I know, viruses infecting aquatic bacteria. Please, correct that.

·       Figure 4: please, italicize the names of the genera.

·       Table 2: please, make it more attractive by removing lines between the rows. Also, it would be better with white background for all table cells.

·       Table 3 and table 4: please, change all cell backgrounds to white.

·       Any significant differences in virome composition related to sex?

Discussion

·       One of the main aspects I think is not enough emphasized is that the study does not gives a complete view of epidermal shark virome but only of phage virome. The authors used a 0.22 µm filter and a DNA extraction kit specialized in phages, what obviously bias their results towards a great proportion of phage reads. This also excludes many larger viruses and RNA viruses. The authors mention this briefly in this section, but I think is something that changes completely the approach of the study, as the results are biased from the very beginning. Thereby, the title of the paper should be reconsidered accordingly.

·       Regarding the proportion of unclassified sequences, the authors may write a small paragraph about the so-called “viral dark matter”, which is very common in high-throughput sequencing of unexplored environments. It would emphasize that further research is needed in the study of viromes of wild animals.

·       Any hypothesis about why the authors were able to find Mimiviruses (around 400 nm diameter) despite using a 0.22 µm filter pore? That has to be clarified.

·       In the paragraph starting with “A majority of the reads from all viromes mapped to bacteriophages”, viral family names must be italicized. The same in the following paragraph.

·       I disagree that the studies on tuna and coral viromes used the same methods as in this study. They extracted all viral RNA and DNA viruses without biases. As aforementioned, the authors extracted only DNA using a phage-specialized kit, what will bias their results (apart from the size of the filter pore). These differences between these two studies and theirs must be mentioned in the text.

·       The reference of the study on coral virome is miscited (is reference no. 43 instead of no. 44).

References

·       All references should follow the same style of citation, either all first letters capitalized or only the first letter capitalized, but not both styles together. Please, unify this along this section.

·       Please, review the italics and the proper use of capital letters of the species names in the titles (for example, Triakis semifasciata in reference no. 20).

Reviewer 2 Report

The authors present a very interesting study of the epidermal virome of sharks which adds to the current knowledge of animal viromes. Epidermal and skin virome studies are still very scarce so this study adds to the layer of this unknown information.

Section 2.1

The authors state that the syringes were filled with seawater and then it was filtered through a tangential flow filter system to remove microbes. Can the authors discuss why this method was appropriate for the collection of the shark epidermal samples over other methods?

Line 115. Can the authors indicate how much time passed since sample collection?

Section 2.2.

Can the author briefly discuss why other techniques such as ultracentrifugation were not part of the protocol to concentrate viruses?

Line 137: Replace ‘Read annotation data was’ to ‘Read annotation data were’

Can the authors indicate in the methods section if a ‘blank control’ was added to the analysis (i.e., extracting and sequencing a sea water sample after tangential flow filtering) to make sure that the shark epidermal viromes were not influenced by the viral communities in the sea water sample? I would think it is expected to have a degree of ‘contamination’ from the sea water, but is there a way to estimate which viruses are from sea water and which viruses are authochtonous to the shark epidermis?

Line 172. Can the authors reference the figure or data showing the presence of cyanophages in the water column sample?

Line 176. Can the authors clarify what percentage of the reads mapped to viruses? Are these numbers the ones shown in Table 1? If so, is 70.5% then the portion of reads that mapped that correspond to phages? The way the sentence is stated reads like most of the reads mapped to the database when according to Table 1 this was not the case.

Also, can the authors indicate in the manuscript what database Hecatomb uses for the virome analysis?

The title may not reflect the true finding of the study, since as it is now it may be suggesting that additional analyses were done to characterize this viral dark matter. Something along the lines of “Shark epidermal viromes are species-specific and dominated by uncharacterized phages” may be more suitable

Reviewer 3 Report

The authors examined the epidermal virome compositions of three shark species:

the genetically and morphologically similar Carcharhinus obscurus and Carcharhinus galapagensis and the outgroup Galeocerdo cuvier. By analyzing the viral metagenomic sequences, the authors identified species-specific virome for these sharks that are dominated by bacteriophages. They also found a relative more similar epidermal virome between C. galapagensis and G. cuvier than to C. obscurus, which is unexpected, suggesting behavioral niche may be a more prominent driver than phylogeny. Overall, this study provided a thorough story describing shark epidermal virome that have been overlooked before and presented intriguing novel findings. Some major and minor points for consideration are listed below.

Major:

1.     The sample size for water column is one. Is it enough to represent the surrounding environmental controls for the sharks? In addition, labeling the water column location in Figure 2A would be helpful too.

2.     Please comment on whether the site of collection on sharks may impact the virome composition and final interpretations.

3.     Line 80-89 described the shapes of shark dermal denticles and physiological features as a part of introduction. However, the author did not really focus on addressing the impacts of dermal denticles on the epidermal virome until the very last part of the discussion. It seems that the content of line 80-89 fit better in the discussion than in the introduction.

4.     Following line 75, it would be better to briefly summarize the key findings of the study to conclude the introduction section.

Minor:

1.     Delete part 0 “How to use the template”

2.     Some of the virus families were italicized, some of them were not. Please correct.

Round 2

Reviewer 1 Report

Only one minor final comment: I would cite the supplementary table somewhere in the main text.

You have done a good job! 

Reviewer 2 Report

The authors have satisfactorily addressed all the comments.